# SPACE GROUP CONSTRAINED CRYSTAL GENERATION

**Rui Jiao**[1,2]***,** **Wenbing Huang**[3,4]†**,** **Yu Liu**[5]**,** **Deli Zhao**[5]**,** **Yang Liu**[1,2]†

[1]Dept. of Comp. Sci. & Tech., Institute for AI, Tsinghua University
[2]Institute for AIR, Tsinghua University
[3]Gaoling School of Artificial Intelligence, Renmin University of China
[4]Beijing Key Laboratory of Big Data Management and Analysis Methods, Beijing, China
[5]Alibaba Group

## ABSTRACT

Crystals are the foundation of numerous scientific and industrial applications. While various learning-based approaches have been proposed for crystal generation, existing methods seldom consider the space group constraint which is crucial in describing the geometry of crystals and closely relevant to many desirable properties. However, considering space group constraint is challenging owing to its diverse and nontrivial forms. In this paper, we reduce the space group constraint into an equivalent formulation that is more tractable to be handcrafted into the generation process. In particular, we translate the space group constraint into two parts: the basis constraint of the invariant logarithmic space of the lattice matrix and the Wyckoff position constraint of the fractional coordinates. Upon the derived constraints, we then propose DiffCSP++, a novel diffusion model that has enhanced a previous work DiffCSP (Jiao et al., 2023) by further taking space group constraint into account. Experiments on several popular datasets verify the benefit of the involvement of the space group constraint, and show that our DiffCSP++ achieves promising performance on crystal structure prediction, ab initio crystal generation and controllable generation with customized space groups.

## 1 INTRODUCTION

Crystal generation represents a critical task in the realm of scientific computation and industrial applications. The ability to accurately and efficiently generate crystal structures opens up avenues for new material discovery and design, thereby having profound implications for various fields, including physics, chemistry, and material science (Liu et al., 2017; Oganov et al., 2019).

Recent advancements in machine learning have paved the way for the application of generative models to this task (Nouira et al., 2018; Hoffmann et al., 2019; Hu et al., 2020; Ren et al., 2021). Among various strategies, diffusion models have been exhibited to be particularly effective in generating realistic and diverse crystal structures (Xie et al., 2021; Jiao et al., 2023). These methods leverage a stochastic process to gradually transform a random initial state into a stable distribution, effectively capturing the complex landscapes of crystal structures.

Despite the success of existing methods, one significant aspect that has been largely overlooked is the consideration of space group symmetry (Hiller, 1986). Space groups play a pivotal role in crystallography, defining the geometry of crystal structures and being intrinsically tied to many properties such as the topological phases (Tang et al., 2019; Chen et al., 2022). However, integrating space group symmetry into diffusion models is a non-trivial task due to the diverse and complex forms of space groups.

In this paper, we supplement this piece by introducing a novel approach that effectively takes space group constraints into account. Our method, termed DiffCSP++, enhances the previous DiffCSP method (Jiao et al., 2023) by translating the space group constraint into a more manageable form, which can be seamlessly integrated into the diffusion process. Our contributions can be summarized as follows:

---

*This work is done when Rui Jiao works as an intern in Alibaba Group.
†Wenbing Huang and Yang Liu are corresponding authors.

- We propose to equivalently interpret the space group constraint into two tractable parts: the basis constraint of the O(3)-invariant logarithmic space of the lattice matrix in § 4.1 and the Wyckoff position constraint of the fractional coordinates in § 4.2, which largely facilitates the incorporation of the space group constraint into the crystal generation process.

- Our method DiffCSP++ separately and simultaneously generates the lattices, fractional coordinates and the atom composition under the reduced form of the space group constraint, through a novel denoising model that is E(3)-invariant.

- Extensive experiments demonstrate that our method not only respects the crucial space group constraints but also achieves promising performance in crystal structure prediction and ab initio crystal generation.

## 2 RELATED WORKS

**Learning-based Crystal Generation.** Data-driven approaches have emerged as a promising direction in the field of crystal generation. These techniques, rather than employing graph-based models, depicted crystals through alternative representations such as voxels voxels (Court et al., 2020; Hoffmann et al., 2019; Noh et al., 2019), distance matrices (Yang et al., 2021; Hu et al., 2020; 2021) or 3D coordinates (Nouira et al., 2018; Kim et al., 2020; Ren et al., 2021). Recently, CDVAE (Xie et al., 2021) combines the VAE backbone with a diffusion-based decoder, and generates the atom types and coordinates on a multi-graph (Xie & Grossman, 2018) built upon predicted lattice parameters. DiffCSP (Jiao et al., 2023) jointly optimizes the lattice matrices and atom coordinates via a diffusion framework. Based on the joint diffusion paradigm, MatterGen (Zeni et al., 2023) applies polar decomposition to represent lattices as O(3)-invariant symmetry matrices, and GemsDiff (Klipfel et al., 2024) projects the lattice matrices onto a decomposed linear vector space. Although these approaches share similar lattice representations with our method, they often overlook the constraints imposed by space groups. Addressing this gap, PGCGM (Zhao et al., 2023) incorporates the affine matrices of the space group as additional input into a Generative Adversarial Network (GAN) model. However, the application of PGCGM is constrained by ternary systems, thus limiting its universality and rendering it inapplicable to all datasets applied in this paper. Besides, PCVAE (Liu et al., 2023) integrates space group constraints to predict lattice parameters using a conditional VAE. In contrast, we impose constraints on the lattice within the logarithmic space, ensuring compatibility with the diffusion-based framework. Moreover, we further specify the Wyckoff position constraints of all atoms, achieving the final goal of structure prediction.

**Diffusion Generative Models.** Diffusion models have been recognized as a powerful generative framework across various domains. Initially gaining traction in the field of computer vision (Ho et al., 2020; Rombach et al., 2021; Ramesh et al., 2022), the versatility of diffusion models has been demonstrated in their application to the generation of small molecules (Xu et al., 2021; Hoogeboom et al., 2021), protein structures (Luo et al., 2022) and crystalline materials (Xie et al., 2021; Jiao et al., 2023). Notably, Chroma (Ingraham et al., 2022) incorporates symmetry conditions into the generation process for protein structures. Different from symmetric proteins, space group constraints require reliable designs for the generation of lattices and special Wyckoff positions, which is mainly discussed in this paper.

## 3 PRELIMINARIES

**Crystal Structures** A crystal structure $\mathcal{M}$ describes the periodic arrangement of atoms in 3D space. The repeating unit is called a *unit cell*, which can be characterized by a triplet, denoted as $(\boldsymbol{A}, \boldsymbol{X}, \boldsymbol{L})$, where $\boldsymbol{A} = [\boldsymbol{a}_1, \boldsymbol{a}_2, ..., \boldsymbol{a}_N] \in \mathbb{R}^{h \times N}$ represents the one-hot representations of atom type, $\boldsymbol{X} = [\boldsymbol{x}_1, \boldsymbol{x}_2, ..., \boldsymbol{x}_N] \in \mathbb{R}^{3 \times N}$ comprises the atoms' Cartesian coordinates, and $\boldsymbol{L} = [\boldsymbol{l}_1, \boldsymbol{l}_2, \boldsymbol{l}_3] \in \mathbb{R}^{3 \times 3}$ is the lattice matrix containing three basic vectors to periodically translate the unit cell to the entire 3D space, which can be extended as $\mathcal{M} := \{(\boldsymbol{a}_i, \boldsymbol{x}'_i) | \boldsymbol{x}'_i \stackrel{\boldsymbol{L}}{=} \boldsymbol{x}_i\}$, where $\boldsymbol{x}'_i \stackrel{\boldsymbol{L}}{=} \boldsymbol{x}_i$ denotes that $\boldsymbol{x}'_i$ is equivalent to $\boldsymbol{x}_i$ if $\boldsymbol{x}'_i$ can be obtained via an integral translation of $\boldsymbol{x}_i$ along the lattices $\boldsymbol{L}$ *i.e.*,

$$\boldsymbol{x}'_i \stackrel{\boldsymbol{L}}{=} \boldsymbol{x}_i \Leftrightarrow \exists \boldsymbol{k}_i \in \mathbb{Z}^{3 \times 1}, \text{s.t. } \boldsymbol{x}'_i = \boldsymbol{x} + \boldsymbol{L}\boldsymbol{k}_i. \tag{1}$$

Apart from the prevalent Cartesian coordinate system, fractional coordinates are also widely applied in crystallography. Given a lattice matrix $\boldsymbol{L} = [\boldsymbol{l}_1, \boldsymbol{l}_2, \boldsymbol{l}_3]$, the fractional coordinate $\boldsymbol{f} = $

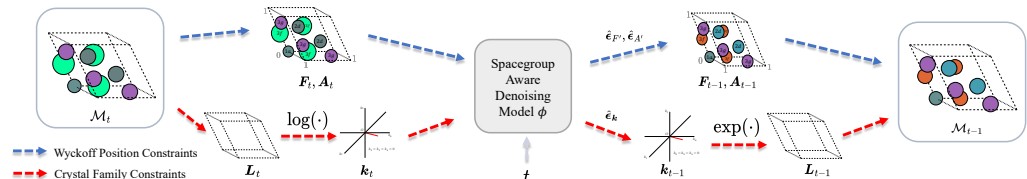

Figure 1: Overview of our proposed DiffCSP++ for the denoising from $\mathcal{M}_t$ to $\mathcal{M}_{t-1}$. We decompose the space group constraints as the crystal family constraints on the lattice matrix (the red dashed line) and the Wyckoff position constraints on each atom (the blue dashed line).

$(f_1, f_2, f_3)^\top \in [0,1)^3$ locates the atom at $\boldsymbol{x} = \sum_{j=1}^{3} f_j \boldsymbol{l}_j$. More generally, given a Cartesian coordinate matrix $\boldsymbol{X} = [\boldsymbol{x}_1, \boldsymbol{x}_2, ..., \boldsymbol{x}_N]$, the corresponding fractional matrix is derived as $\boldsymbol{F} = \boldsymbol{L}^{-1}\boldsymbol{X}$.

**Space Group** The concept of space group is used to describe the inherent symmetry of a crystal structure. Given a transformation $g \in \mathrm{E}(3)$, we define the transformation of the coordinate matrix $\boldsymbol{X}$ as $g \cdot \boldsymbol{X}$ which is implemented as $g \cdot \boldsymbol{X} \coloneqq \boldsymbol{OX} + \boldsymbol{t1}^\top$ for a orthogonal matrix $\boldsymbol{O} \in \mathrm{O}(3)$, a translation vector $\boldsymbol{t} \in \mathbb{R}^3$ and a 3-dimensional all-ones vector $\boldsymbol{1}$. If $g$ lets $\mathcal{M}$ invariant, that is $g \cdot \mathcal{M} \coloneqq \{(\boldsymbol{a}_i, g \cdot \boldsymbol{x}_i)\} = \mathcal{M}$ (note that the symbol "=" here refers to the equivalence between sets), $\mathcal{M}$ is recognized to be symmetric with respect to $g$. The space group symmetry $g \cdot \mathcal{M} = \mathcal{M}$ can also be depicted by checking how the atoms are transformed. Specifically, for each transformation $g \in G(\mathcal{M})$, there exists a permutation matrix $\boldsymbol{P}_g \in \{0,1\}^{N \times N}$ that maps each atom to its corresponding symmetric point:

$$\boldsymbol{A} = \boldsymbol{AP}_g, \quad g \cdot \boldsymbol{X} \stackrel{L}{=} \boldsymbol{XP}_g, \tag{2}$$

The set of all possible symmetric transformations of $\mathcal{M}$ constitutes a space group $G(\mathcal{M}) = \{g \in \mathrm{E}(3) | g \cdot \mathcal{M} = \mathcal{M}\}$. Owing to the periodic nature of crystals, the size of $G(\mathcal{M})$ is finite, and the total count of different space groups is finite as well. It has been conclusively demonstrated that there are 230 kinds of space groups for all crystals.

**Task Definition** We focus on generating space group-constrained crystals by learning a conditional distribution $p(\mathcal{M}|G)$, where $G$ is the given space group with size $|G| = m$. Most previous works (Xie et al., 2021; Jiao et al., 2023) derive $p(\mathcal{M})$ without $G$ and they usually apply E(3)-equivariant generative models to implement $p(\mathcal{M})$ to eliminate the influence by the choice of the coordinate systems. In this paper, the O(3) equivariance is no longer required as both $\boldsymbol{L}$ and $\boldsymbol{X}$ will be embedded to invariant quantities, which will be introduced in § 4.1. The translation invariance and periodicity will be maintained under the Fourier representation of the fractional coordinates, which will be shown in § 4.4.

## 4 THE PROPOSED METHOD: DIFFCSP++

It is nontrivial to exactly involve the constraint of Eq. 2 into existing generative models, due to the various types of the space group constraints. In this section, we will reduce the space group constraint from two aspects: the invariant representation of constrained lattice matrices in § 4.1 and the Wyckoff positions of fractional coordinates in § 4.2, which will be tractably and inherently maintained during our proposed diffusion process in § 4.3.

### 4.1 INVARIANT REPRESENTATION OF LATTICE MATRICES

The lattice matrix $\boldsymbol{L} \in \mathbb{R}^{3 \times 3}$ determines the shape of the unit cell. If the determinant (namely the volume) of $\boldsymbol{L}$ is meaningful: $\det(\boldsymbol{L}) > 0$, then the lattice matrix is invertible and we have the following decomposition.

**Proposition 1** (Polar Decomposition (Hall, 2013)). *An invertible matrix $\boldsymbol{L} \in \mathbb{R}^{3 \times 3}$ can be uniquely decomposed into $\boldsymbol{L} = \boldsymbol{Q}\exp(\boldsymbol{S})$, where $\boldsymbol{Q} \in \mathbb{R}^{3 \times 3}$ is an orthogonal matrix, $\boldsymbol{S} \in \mathbb{R}^{3 \times 3}$ is a symmetric matrix and $\exp(\boldsymbol{S}) = \sum_{n=0}^{\infty} \frac{\boldsymbol{S}^n}{n!}$ defines the exponential mapping of $\boldsymbol{S}$.*

The above proposition indicates that $\boldsymbol{L}$ can be uniquely represented by a symmetric matrix $\boldsymbol{S}$. Moreover, any O(3) transformation of $\boldsymbol{L}$ leaves $\boldsymbol{S}$ unchanged, as the transformation will be reflected by

$\boldsymbol{Q}$. We are able to find 6 bases of the space of symmetric matrices, *e.g.*,

$$\boldsymbol{B}_1 = \begin{pmatrix} 0 & 1 & 0 \\ 1 & 0 & 0 \\ 0 & 0 & 0 \end{pmatrix}, \boldsymbol{B}_2 = \begin{pmatrix} 0 & 0 & 1 \\ 0 & 0 & 0 \\ 1 & 0 & 0 \end{pmatrix}, \boldsymbol{B}_3 = \begin{pmatrix} 0 & 0 & 0 \\ 0 & 0 & 1 \\ 0 & 1 & 0 \end{pmatrix},$$

$$\boldsymbol{B}_4 = \begin{pmatrix} 1 & 0 & 0 \\ 0 & -1 & 0 \\ 0 & 0 & 0 \end{pmatrix}, \boldsymbol{B}_5 = \begin{pmatrix} 1 & 0 & 0 \\ 0 & 1 & 0 \\ 0 & 0 & -2 \end{pmatrix}, \boldsymbol{B}_6 = \begin{pmatrix} 1 & 0 & 0 \\ 0 & 1 & 0 \\ 0 & 0 & 1 \end{pmatrix}.$$

Each symmetric matrix can be expanded via the above symmetric bases as stated below.

**Proposition 2.** $\forall \boldsymbol{S} \in \mathbb{R}^{3 \times 3}, \boldsymbol{S} = \boldsymbol{S}^\top, \exists \boldsymbol{k} = (k_1, \cdots, k_6), s.t. \boldsymbol{S} = \sum_{i=1}^{6} k_i \boldsymbol{B}_i.$

By joining the conclusions of Propositions 1 and 2, it is clear to find that $\boldsymbol{L}$ is determined by the values of $k_i$. Therefore, we are able to choose different combinations of the symmetric bases to reflect the space group constraint acting on $\boldsymbol{L}$. Actually, the total 230 space groups are classified into 6 crystal families, determining the shape of $\boldsymbol{L}$. After careful derivations (provided in Appendix A.3), the correspondence between the crystal families and the values of $k_i$ is given in the following table. With such a table, when we want to generate the lattice restricted by a given space group, we can first retrieve the crystal family and then enforce the corresponding constraint on $k_i$ during generation.

Table 1: Relationship between the lattice shape and the constraint of the symmetric bases, where $a, b, c$ and $\alpha, \beta, \gamma$ denote the lengths and angles of the lattice bases, respectively.

| Crystal Family | Space Group No. | Lattice Shape | Constraint of Symmetric Bases |
|---|---|---|---|
| Triclinic | $1 \sim 2$ | No Constraint | No Constraint |
| Monoclinic | $3 \sim 15$ | $\alpha = \gamma = 90°$ | $k_1 = k_3 = 0$ |
| Orthorhombic | $16 \sim 74$ | $\alpha = \beta = \gamma = 90°$ | $k_1 = k_2 = k_3 = 0$ |
| Tetragonal | $75 \sim 142$ | $\alpha = \beta = \gamma = 90°$ 
 $a = b$ | $k_1 = k_2 = k_3 = 0$ 
 $k_4 = 0$ |
| Hexagonal | $143 \sim 194$ | $\alpha = \beta = 90°, \gamma = 120°$ 
 $a = b$ | $k_2 = k_3 = 0, k_1 = -log(3)/4$ 
 $k_4 = 0$ |
| Cubic | $195 \sim 230$ | $\alpha = \beta = \gamma = 90°$ 
 $a = b = c$ | $k_1 = k_2 = k_3 = 0$ 
 $k_4 = k_5 = 0$ |

## 4.2 Wyckoff Positions of Fractional Coordinates

As shown in Eq. 2, each transformation $g \in G$ is associated with a permutation matrix $\boldsymbol{P}_g$. Considering atom $i$ for example, it will be transformed to a symmetric and equivalent point $j$, if $\boldsymbol{P}_g[i, j] = 1, i \neq j$, where $\boldsymbol{P}_g[i, j]$ returns the element of the $i$-th row and $j$-th column. Under some particular transformation $g$, the atom $s$ will be transformed to itself, implying that $\boldsymbol{P}_g[i, i] = 1$. Such transformations that leave $i$ invariant comprise the *site symmetry group*, defined as $G_i = \{g \in G | g \cdot \boldsymbol{x}_i \overset{\boldsymbol{L}}{=} \boldsymbol{x}_i\} \subseteq G$. Now, we introduce the notion of *Wyckoff position* that is useful in crystallography. For atom $i$, it shares the same Wyckoff position with atoms owning the conjugate site symmetry groups of $G_i$. There could be multiple types of Wyckoff positions in a unit cell.

Wyckoff positions can also be represented by the fractional coordinate system. Given a crystal structure with $N$ atoms belonging to $N'$ Wyckoff positions, we denote that each Wyckoff position contains $n_s$ atoms satisfying $1 \leq s \leq N'$. The symbol $n_s$ is named as the multiplicity of the $s$-th Wyckoff position and maintains $\sum_{s=1}^{N'} n_s = N$. In the following sections, we denote $\boldsymbol{a}'_s, \boldsymbol{f}'_s$ as the atom type and basic coordinate of the $s$-th Wyckoff position, and $\boldsymbol{a}_{s_i}, \boldsymbol{f}_{s_i}$ as the atom type and fractional coordinate of atom $s_i$ in the $s$-th Wyckoff position. A type of Wyckoff positions is formulated as a list of transformation pairs $\{(\boldsymbol{R}_{s_i}, \boldsymbol{t}_{s_i})\}_{i=1}^{n_s}$ that project the basic coordinate $\boldsymbol{f}'_s$ to all equivalent positions $\{\boldsymbol{R}_{s_i} \boldsymbol{f}'_s + \boldsymbol{t}_{s_i}\}_{i=1}^{n_s}$. Figure 2 illustrates an example with $N' = 7$.

Since Wyckoff positions are inherently determined by the space group, we can realize the space group constraint by restricting the coordinates of the atoms that are located in the same set of the Wyckoff positions during crystal generation, which will be presented in detail in the next subsection.

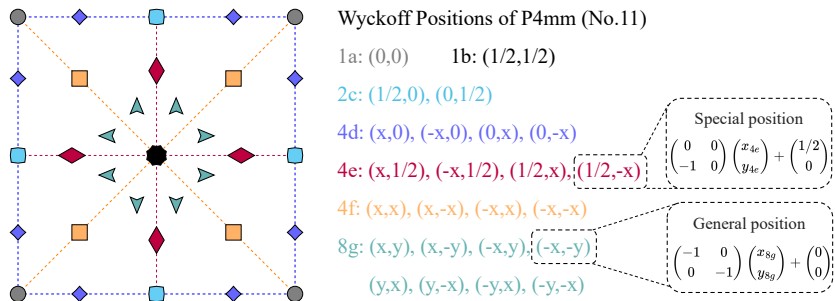

Figure 2: Inspired by PyXtal (Fredericks et al., 2021), we utilize the 2D plain group P4mm as a toy example to demonstrate the concept of Wyckoff position. An asymmetric triangle is copied eight times to construct the square unit cell, and the general Wyckoff position ($8g^1$) has a multiplicity of eight. The other Wyckoff positions are restricted to certain subspaces. For instance, position 4e is constrained by the red dashed lines.

### 4.3 DIFFUSION UNDER SPACE GROUP CONSTRAINTS

To tackle the crystal generation problem, we utilize diffusion models to jointly generate the lattice matrix $L$, the fractional coordinates $F$ and the atom types $A$ under the specific space group constraint. We detail the forward diffusion process and the backward generation process of the three key components as follows.

**Diffusion on $L$.** As mentioned in 4.1, the lattice matrix $L$ can be uniquely represented by an O(3)-invariant coefficient vector $k$. Hence, we directly design the diffusion process on $k$, and the forward probability of time step $t$ is given by

$$q(\boldsymbol{k}_t|\boldsymbol{k}_0) = \mathcal{N}\Big(\boldsymbol{k}_t|\sqrt{\bar{\alpha}_t}\boldsymbol{k}_0, (1-\bar{\alpha}_t)\boldsymbol{I}\Big), \tag{3}$$

where $\bar{\alpha}_t = \prod_{s=1}^{t}(1-\beta_t)$, and $\beta_t \in (0,1)$ determines the variance of each diffusion step, controlled by the cosine scheduler proposed in Nichol & Dhariwal (2021).

Starting from the normal prior $\boldsymbol{k}_T \sim \mathcal{N}(0, \boldsymbol{I})$, the corresponding generation process is designed as

$$p(\boldsymbol{k}_{t-1}|\mathcal{M}_t) = \mathcal{N}(\boldsymbol{k}_{t-1}|\mu_{\boldsymbol{k}}(\mathcal{M}_t), \beta_t\frac{1-\bar{\alpha}_{t-1}}{1-\bar{\alpha}_t}\boldsymbol{I}), \tag{4}$$

where $\mu_{\boldsymbol{k}}(\mathcal{M}_t) = \frac{1}{\sqrt{\alpha_t}}\Big(\boldsymbol{k}_t - \frac{\beta_t}{\sqrt{1-\bar{\alpha}_t}}\hat{\boldsymbol{\epsilon}}_{\boldsymbol{k}}(\mathcal{M}_t, t)\Big)$, and the term $\hat{\boldsymbol{\epsilon}}_{\boldsymbol{k}}(\mathcal{M}_t, t)$ is predicted by the denoising model $\phi(\mathcal{M}_t, t)$.

To confine the structure under a desired space group constraint, we only diffuse and generate the unconstrained dimensions of $k$ while preserving the constrained value $k_0$ as outlined in Table 1. To optimize the denoising term $\hat{\boldsymbol{\epsilon}}_{\boldsymbol{k}}(\mathcal{M}_t, t)$, we first sample $\boldsymbol{\epsilon}_{\boldsymbol{k}} \sim \mathcal{N}(0, \boldsymbol{I})$ and reparameterize $\boldsymbol{k}_t$ as $\boldsymbol{k}_t = \boldsymbol{m}\odot(\sqrt{\bar{\alpha}_t}\boldsymbol{k}_0 + \sqrt{1-\bar{\alpha}_t}\boldsymbol{\epsilon}_{\boldsymbol{k}}) + (1-\boldsymbol{m})\odot\boldsymbol{k}_0$, where the mask is given by $\boldsymbol{m} \in \{0,1\}^6$, $m_i = 1$ indicates the $i$-th basis is unconstrained, and $\odot$ is the element-wise multiplication. The objective on $k$ is finally computed by

$$\mathcal{L}_{\boldsymbol{k}} = \mathbb{E}_{\boldsymbol{\epsilon}_{\boldsymbol{k}}\sim\mathcal{N}(0,\boldsymbol{I}), t\sim\mathcal{U}(1,T)}[\|\boldsymbol{m}\odot\boldsymbol{\epsilon}_{\boldsymbol{k}} - \hat{\boldsymbol{\epsilon}}_{\boldsymbol{L}}(\mathcal{M}_t, t)\|_2^2]. \tag{5}$$

**Diffusion on $F$.** In a unit cell, the fractional coordinates $F \in \mathbb{R}^{3\times N}$ of $N$ atoms can be arranged as the Wyckoff positions of $N'$ basic fractional coordinates $F' \in \mathbb{R}^{3\times N'}$. Hence we only focus on the generation of $F'$, and its forward process is conducted via the Wrapped Normal distribution following Jiao et al. (2023) to maintain periodic translation invariance:

$$q(\boldsymbol{F}'_t|\boldsymbol{F}'_0) = \mathcal{N}_w\Big(\boldsymbol{F}'_t|\boldsymbol{F}'_0, \sigma_t^2\boldsymbol{I}\Big), \tag{6}$$

---

[1]In practical implementation, Wyckoff positions are identified by a combination of a number and a letter, where the number is the multiplicity, and the letter is to distinguish the Wyckoff position type in a dictionary order corresponding to the ascending order of the multiplicity.

The forward sampling can be implemented as $\boldsymbol{F}'_t = w(\boldsymbol{F}'_0 + \sigma_t \boldsymbol{\epsilon}_{\boldsymbol{F}'}(\mathcal{M}_t, t))$, where $w(\cdot)$ retains the fractional part of the input. For the backward process, we first acquire $\boldsymbol{F}'_T$ from the uniform initialization, and sample $\boldsymbol{F}'_0$ via the predictor-corrector sampler with the denoising term $\hat{\boldsymbol{\epsilon}}_{\boldsymbol{F}'}(\mathcal{M}_t, t)$ output by the model $\phi(\mathcal{M}_t, t)$.

Note that the basic coordinates do not always have 3 degrees of freedom. The transformation matrice $\boldsymbol{R}_{s_i}$ could be singular, resulting in Wyckoff positions located in specific planes, axes, or even reduced to fixed points. Hence we project the noise term $\boldsymbol{\epsilon}_{\boldsymbol{F}'}$ onto the constrained subspaces via the least square method as $\boldsymbol{\epsilon}'_{\boldsymbol{F}'}[:, s] = \boldsymbol{R}^\dagger_{s_0} \boldsymbol{\epsilon}_{\boldsymbol{F}'}[:, s]$, where $\boldsymbol{R}^\dagger_i$ is the pseudo-inverse of $\boldsymbol{R}_{s_0}$. The training objective on $\boldsymbol{F}'$ is

$$\mathcal{L}_{\boldsymbol{F}'} = \mathbb{E}_{\boldsymbol{F}'_t \sim q'(\boldsymbol{F}'_t | \boldsymbol{F}'_0), t \sim \mathcal{U}(1,T)} \big[ \lambda_t \| \nabla_{\boldsymbol{F}'_t} \log q'(\boldsymbol{F}'_t | \boldsymbol{F}'_0) - \hat{\boldsymbol{\epsilon}}_{\boldsymbol{F}'}(\mathcal{M}_t, t) \|^2_2 \big], \tag{7}$$

where $\lambda_t = \mathbb{E}^{-1}\big[\|\nabla \log \mathcal{N}_w(0, \sigma_t^2)\|^2_2\big]$ is the pre-computed weight, and $q'$ is the projected distribution of $q$ induced by $\boldsymbol{\epsilon}'_{\boldsymbol{F}'}$. Further details are provided in Appendix.

**Diffusion on $\boldsymbol{A}$.** Since the atom types $\boldsymbol{A}$ remain consistent with the Wyckoff positions, we can also only focus on the basic atoms $\boldsymbol{A}' \subseteq \boldsymbol{A}$. Considering $\boldsymbol{A}' \in \mathbb{R}^{h \times N'}$ as the one-hot continues representation, we apply the standard DDPM-based method by specifying the forward process as

$$q(\boldsymbol{A}'_t | \boldsymbol{A}'_0) = \mathcal{N}\Big(\boldsymbol{A}'_t | \sqrt{\bar{\alpha}_t} \boldsymbol{A}'_0, (1 - \bar{\alpha}_t)\boldsymbol{I}\Big). \tag{8}$$

And the backward process is defined as

$$p(\boldsymbol{A}'_{t-1} | \mathcal{M}_t) = \mathcal{N}(\boldsymbol{A}'_{t-1} | \mu_{\boldsymbol{A}'}(\mathcal{M}_t), \beta_t \frac{1 - \bar{\alpha}_{t-1}}{1 - \bar{\alpha}_t}(\mathcal{M}_t)\boldsymbol{I}), \tag{9}$$

where $\mu_{\boldsymbol{A}'}(\mathcal{M}_t)$ is similar to $\mu_{\boldsymbol{k}}(\mathcal{M}_t)$ in Eq. 4. The denoising term $\hat{\boldsymbol{\epsilon}}_{\boldsymbol{A}'}(\mathcal{M}_t, t) \in \mathbb{R}^{h \times N'}$ is predicted by the model $\phi(\mathcal{M}_t, t)$.

The training objective is

$$\mathcal{L}_{\boldsymbol{A}'} = \mathbb{E}_{\boldsymbol{\epsilon}_{\boldsymbol{A}_s} \sim \mathcal{N}(0, \boldsymbol{I}), t \sim \mathcal{U}(1,T)} \big[ \| \boldsymbol{\epsilon}_{\boldsymbol{A}_s} - \hat{\boldsymbol{\epsilon}}_{\boldsymbol{A}_s}(\mathcal{M}_t, t) \|^2_2 \big]. \tag{10}$$

The entire objective for training the joint diffusion model of $\mathcal{M}$ is combined as

$$\mathcal{L}_{\mathcal{M}} = \lambda_{\boldsymbol{k}} \mathcal{L}_{\boldsymbol{k}} + \lambda_{\boldsymbol{F}'} \mathcal{L}_{\boldsymbol{F}'} + \lambda_{\boldsymbol{A}'} \mathcal{L}_{\boldsymbol{A}'}. \tag{11}$$

## 4.4 Denoising Model

In this subsection, we introduce the specific design of the denoising model $\phi(\mathcal{M}_t, t)$ to obtain the three denoising terms $\hat{\boldsymbol{\epsilon}}_{\boldsymbol{k}}, \hat{\boldsymbol{\epsilon}}_{\boldsymbol{F}'}, \hat{\boldsymbol{\epsilon}}_{\boldsymbol{A}'}$ under the space group constraint, with the detailed architecture illustrated in Figure 4 at Appendix B.1. We omit the subscript $t$ in this subsection for brevity.

We first fuse the atom embeddings $f_{\text{atom}}(\boldsymbol{A})$ and the sinusoidal time embedding $f_{\text{time}}(t)$ to acquire the input node features $\boldsymbol{H} = \varphi_{\text{in}}(f_{\text{atom}}(\boldsymbol{A}), f_{\text{time}}(t))$, where $\varphi_{\text{in}}$ is an MLP. The message passing from node $j$ to $i$ in the $l$-th layer is designed as Eq. (12-13), where $\varphi_m$ and $\varphi_h$ are MLPs, and $\psi_{\text{FT}} : (-1,1)^3 \to [-1,1]^{3 \times K}$ is the Fourier transformation with $K$ bases to periodically embed the relative fractional coordinate $\boldsymbol{f}_j - \boldsymbol{f}_i$. Note that here we apply $\boldsymbol{k}$ in Eq. (12) as the unique O(3)-invariant representation of $\boldsymbol{L}$ instead of the inner product $\boldsymbol{L}^\top \boldsymbol{L}$ in DiffCSP (Jiao et al., 2023), and its reliability is validated in § 5.4. After $L$ layers of message passing, we get the invariant graph- and node-level denoising terms as Eq. (14-15), where $\varphi_{\boldsymbol{k}}, \varphi_{\boldsymbol{F}}, \varphi_{\boldsymbol{A}}$ are MLPs. To align with the constrained diffusion framework proposed in § 4.3, the denosing terms are re-

$$\boldsymbol{m}^{(l)}_{ij} = \varphi_m(\boldsymbol{h}^{(l-1)}_i, \boldsymbol{h}^{(l-1)}_j, \boldsymbol{k}, \psi_{\text{FT}}(\boldsymbol{f}_j - \boldsymbol{f}_i)), \tag{12}$$

$$\boldsymbol{h}^{(l)}_i = \boldsymbol{h}^{(l-1)}_i + \varphi_h(\boldsymbol{h}^{(l-1)}_i, \sum_{j=1}^N \boldsymbol{m}^{(l)}_{ij}), \tag{13}$$

$$\hat{\boldsymbol{\epsilon}}_{\boldsymbol{k}, \text{unconstrained}} = \varphi_{\boldsymbol{k}}\Big(\frac{1}{N}\sum_{i=1}^N \boldsymbol{h}^{(L)}_i\Big), \tag{14}$$

$$\hat{\boldsymbol{\epsilon}}_{\boldsymbol{F}}[:, i], \hat{\boldsymbol{\epsilon}}_{\boldsymbol{A}}[:, i] = \varphi_{\boldsymbol{F}}(\boldsymbol{h}^{(L)}_i), \varphi_{\boldsymbol{A}}(\boldsymbol{h}^{(L)}_i), \tag{15}$$

$$\hat{\boldsymbol{\epsilon}}_{\boldsymbol{k}} = \boldsymbol{m} \odot \hat{\boldsymbol{\epsilon}}_{\boldsymbol{k}, \text{unconstrained}}, \tag{16}$$

$$\hat{\boldsymbol{\epsilon}}_{\boldsymbol{F}'} = \text{WyckoffMean}(\hat{\boldsymbol{\epsilon}'_{\boldsymbol{F}}}), \tag{17}$$

$$\hat{\boldsymbol{\epsilon}}_{\boldsymbol{A}'} = \text{WyckoffMean}(\hat{\boldsymbol{\epsilon}}_{\boldsymbol{A}}), \tag{18}$$

quired to maintain the space group constraints, which is not considered in the original DiffCSP. The constrained denoising terms are finally projected as Eq. (16-18), where $\hat{\boldsymbol{\epsilon}}'_{\boldsymbol{F}}[:, i] = \boldsymbol{R}^\dagger_i \hat{\boldsymbol{\epsilon}}_{\boldsymbol{F}}[:, i]$ is the projected denoising term towards the subspace of the Wyckoff positions, and WyckoffMean computes the average of atoms belonging to the same Wyckoff position.

Table 2: Results on crystal structure prediction task. MR stands for Match Rate.

| | Perov-5 | | MP-20 | | MPTS-52 | |
| --- | --- | --- | --- | --- | --- | --- |
| | MR (%) | RMSE | MR (%) | RMSE | MR (%) | RMSE |
| RS | 36.56 | 0.0886 | 11.49 | 0.2822 | 2.68 | 0.3444 |
| BO | 55.09 | 0.2037 | 12.68 | 0.2816 | 6.69 | 0.3444 |
| PSO | 21.88 | 0.0844 | 4.35 | 0.1670 | 1.09 | 0.2390 |
| P-cG-SchNet (Gebauer et al., 2022) | 48.22 | 0.4179 | 15.39 | 0.3762 | 3.67 | 0.4115 |
| CDVAE (Xie et al., 2021) | 45.31 | 0.1138 | 33.90 | 0.1045 | 5.34 | 0.2106 |
| DiffCSP (Jiao et al., 2023) | 52.02 | **0.0760** | 51.49 | 0.0631 | 12.19 | 0.1786 |
| DiffCSP++ (w/ CSPML) | **52.17** | 0.0841 | **70.58** | **0.0272** | **37.17** | **0.0676** |
| DiffCSP++ (w/ GT) | 98.44 | 0.0430 | 80.27 | 0.0295 | 46.29 | 0.0896 |

## 5 EXPERIMENTS

### 5.1 SETUP

In this section, we evaluate our method over various tasks. We demonstrate the capability and explore the potential upper limits on the crystal structure prediction task in § 5.2. Additionally, we present the remarkable performance achieved in the ab initio generation task in § 5.3. We further provide adequate analysis in § 5.4.

**Datasets.** We evaluate our method on four datasets with different data distributions. **Perov-5** (Castelli et al., 2012) encompasses 18,928 perovskite crystals with similar structures but distinct compositions. Each structure has precisely 5 atoms in a unit cell. **Carbon-24** (Pickard, 2020) comprises 10,153 carbon crystals. All the crystals share only one element, Carbon, while exhibiting diverse structures containing $6 \sim 24$ atoms within a unit cell. **MP-20** (Jain et al., 2013) contains 45,231 materials sourced from Material Projects with diverse compositions and structures. These materials represent the majority of experimentally generated crystals, each consisting of no more than 20 atoms in a unit cell. **MPTS-52** serves as a more challenging extension of MP-20, consisting of 40,476 structures with unit cells containing up to 52 atoms. For Perov-5, Carbon-24 and MP-20, we follow the 60-20-20 split with previous works (Xie et al., 2021). For MPTS-52, we perform a chronological split, allocating 27,380/5,000/8,096 crystals for training/validation/testing.

**Tasks.** We focus on two major tasks attainable through our method. **Crystal Structure Prediction (CSP)** aims at predicting the structure of a crystal based on its composition. **Ab Initio Generation** requires generating crystals with valid compositions and stable structures. We conduct the CSP experiments on Perov-5, MP-20, and MPTS-52, as the structures of carbon crystals vary diversely, and it is not reasonable to match the generated samples with one specific reference on Carbon-24. The comparison for the generation task is carried out using Perov-5, Carbon-24, and MP-20, aligning with previous works.

### 5.2 CRYSTAL STRUCTURE PREDICTION

To adapt our method to the CSP task, we keep the atom types unchanged during the training and generation stages. Moreover, the proposed DiffCSP++ requires the provision of the space group and the Wyckoff positions of all atoms during the generation process. To address this requirement, we employ two distinct approaches to obtain these essential conditions.

For the first version of our method, we select the space group of the Ground-Truth (GT) data as input. However, it's worth noting that these conditions are typically unavailable in real-world scenarios. Instead, we also implement our method with CSPML (Kusaba et al., 2022), a metric learning technique designed to select templates for the prediction of new structures. Given a composition, we first identify the composition in the training set that exhibits the highest similarity. Subsequently, we employ the corresponding structure as a template and refine it using DiffCSP++ after proper element substitution. We provide more details in Appendix B.2.

For evaluation, we match the predicted sample with the ground truth structure. For each composition within the testing set, we generate one structure and the match is determined by the StructureMatcher class in pymatgen (Ong et al., 2013) with thresholds stol=0.5, angle_tol=10, ltol=0.3, in accordance

Table 3: Results on ab initio generation task. The results of baseline methods are from Xie et al. (2021); Jiao et al. (2023).

| Data | Method | Validity (%) ↑ | | Coverage (%) ↑ | | Property ↓ | | |
|------|--------|------|------|------|------|------|------|------|
| | | Struc. | Comp. | COV-R | COV-P | $d_\rho$ | $d_E$ | $d_{elem}$ |
| Perov-5 | FTCP (Ren et al., 2021) | 0.24 | 54.24 | 0.00 | 0.00 | 10.27 | 156.0 | 0.6297 |
| | Cond-DFC-VAE (Court et al., 2020) | 73.60 | 82.95 | 73.92 | 10.13 | 2.268 | 4.111 | 0.8373 |
| | G-SchNet (Gebauer et al., 2019) | 99.92 | 98.79 | 0.18 | 0.23 | 1.625 | 4.746 | 0.0368 |
| | P-G-SchNet (Gebauer et al., 2019) | 79.63 | **99.13** | 0.37 | 0.25 | 0.2755 | 1.388 | 0.4552 |
| | CDVAE (Xie et al., 2021) | **100.0** | 98.59 | 99.45 | 98.46 | 0.1258 | 0.0264 | 0.0628 |
| | DiffCSP (Jiao et al., 2023) | **100.0** | 98.85 | **99.74** | 98.27 | 0.1110 | **0.0263** | 0.0128 |
| | DiffCSP++ | **100.0** | 98.77 | 99.60 | **98.80** | **0.0661** | 0.0405 | **0.0040** |
| Carbon-24 | FTCP (Ren et al., 2021) | 0.08 | – | 0.00 | 0.00 | 5.206 | 19.05 | – |
| | G-SchNet (Gebauer et al., 2019) | 99.94 | – | 0.00 | 0.00 | 0.9427 | 1.320 | – |
| | P-G-SchNet (Gebauer et al., 2019) | 48.39 | – | 0.00 | 0.00 | 1.533 | 134.7 | – |
| | CDVAE (Xie et al., 2021) | **100.0** | – | 99.80 | 83.08 | 0.1407 | 0.2850 | – |
| | DiffCSP (Jiao et al., 2023) | **100.0** | – | 99.90 | **97.27** | 0.0805 | **0.0820** | – |
| | DiffCSP++ | 99.99 | – | **100.0** | 88.28 | **0.0307** | 0.0935 | – |
| MP-20 | FTCP (Ren et al., 2021) | 1.55 | 48.37 | 4.72 | 0.09 | 23.71 | 160.9 | 0.7363 |
| | G-SchNet (Gebauer et al., 2019) | 99.65 | 75.96 | 38.33 | 99.57 | 3.034 | 42.09 | 0.6411 |
| | P-G-SchNet (Gebauer et al., 2019) | 77.51 | 76.40 | 41.93 | 99.74 | 4.04 | 2.448 | 0.6234 |
| | CDVAE (Xie et al., 2021) | **100.0** | **86.70** | 99.15 | 99.49 | 0.6875 | 0.2778 | 1.432 |
| | DiffCSP (Jiao et al., 2023) | **100.0** | 83.25 | 99.71 | **99.76** | 0.3502 | 0.1247 | **0.3398** |
| | DiffCSP++ | 99.94 | 85.12 | **99.73** | 99.59 | **0.2351** | **0.0574** | 0.3749 |

with previous setups. The match rate represents the ratio of matched structures relative to the total number within the testing set, and the RMSD is averaged over the matched pairs, and normalized by $\sqrt[3]{V/N}$ where $V$ is the volume of the lattice.

We compare our methods with two lines of baselines. The first line is the optimization-based methods (Cheng et al., 2022) including Random Search (**RS**), Bayesian Optimization (**BO**), and Particle Swarm Optimization (**PSO**). The second line considers three types of generative methods. **P-cG-SchNet** (Gebauer et al., 2022) is an autoregressive model taking the composition as the condition. **CDVAE** (Xie et al., 2021) proposes a VAE framework that first predicts the invariant lattice parameters and then generates the atom types and coordinates via a score-based decoder. **DiffCSP** (Jiao et al., 2023) jointly generates the lattices and atom coordinates. All the generative methods do not consider the space group constraints.

The results are shown in Table 2, where we provide the performance of the templates mined by CSPML and directly from GT. We have the following observations. **1.** DiffCSP++, when equipped with GT conditions, demonstrates a remarkable superiority over other methods. This indicates that incorporating space group symmetries into the generation framework significantly enhances its ability to predict more precise structures. **2.** When combined with CSPML templates, our method continues to surpass baseline methods. Given that the ground truth (GT) space groups are not accessible in real-world CSP scenes, our method offers a practical solution for predicting structures with high space group symmetry. **3.** Notably, there remains a gap between match rates under space group conditions derived from mined templates and those from GT conditions (70.58% vs. 80.27% on MP-20). This suggests that an improved template-finding algorithm could potentially enhance performance, which we leave for further studies.

## 5.3 Ab Initio Generation

For each dataset, we first sample 10,000 structures from the training set with replacement as templates, and conduct the ab initio generation on the extracted templates. We focus on three lines of metrics for evaluation. **Validity.** We requires both the structures and the compositions of the generated samples are valid. The structural valid rate is the ratio of the samples with the minimal pairwise distance larger than 0.5Å, while the compositional valid rate is the percentage of samples under valence equilibrium solved by SMACT (Davies et al., 2019). **Coverage.** The coverage recall (COV-R) and precision (COV-P) calculate the percentage of the crystals in the testing set and that in generated samples matched with each other within a fingerprint distance threshold. **Property statistics.** We calculate three Wasserstein distances between the generated and testing structures, specifically focusing on density, formation energy, and the number of elements (Xie et al., 2021), denoted as $d_\rho$, $d_E$, and $d_{elem}$ respectively. To execute this evaluation, we apply these validity and

coverage metrics to all 10,000 generated samples, and the property metrics are computed on a subset of 1,000 valid samples.

We compare our method with previous generative methods **FTCP** (Ren et al., 2021), **Cond-DFC-VAE** (Court et al., 2020), **G-SchNet** (Gebauer et al., 2019), **P-G-SchNet**, **CDVAE** (Xie et al., 2021) and **DiffCSP** (Jiao et al., 2023). Table 3 depicts the results. We find that our method yields comparable performance on validity and coverage metrics, while showcasing a substantial superiority over the baselines when it comes to property statistics, indicating that the inclusion of space group constraints contributes to the model's ability to generate more realistic crystals, especially for complex structures like in MP-20.

## 5.4 ANALYSIS

In this subsection, we discuss the influence of the key components in our proposed framework.

**Invariant Lattice Representation.** In this work, we substitute the coefficient vector $k$ for the inner product term $L^\top L$ in DiffCSP (Jiao et al., 2023) to serve as the O(3)-invariant representation of the lattice matrix. To assess the impact of this modification, we adapt the diffusion process and representation from $L$ to $k$ in DiffCSP, without imposing extra space group constraints. This variant is denoted as DiffCSP-$k$. The performance of DiffCSP-$k$ is substantiated by the results in Table 4 as being on par with the original DiffCSP, validating $k$ as a dependable invariant representation of the lattice matrix.

Table 4: Ablation studies.

| MP-20 | MR (%) | RMSE |
|---|---|---|
| *Invariant Lattice Representation* | | |
| DiffCSP | 51.49 | 0.0631 |
| DiffCSP-$k$ | 50.76 | 0.0608 |
| *Pre-Average vs. Post-Average* | | |
| DiffCSP++ (Pre) | 78.75 | 0.0355 |
| DiffCSP++ (Post) | 80.27 | 0.0295 |

**Pre-Average vs. Post-Average.** In Eq. (17), we average the denoising outputs on $F$ to the base nodes for each Wyckoff position, and calculate the losses on the base node in Eq. (7). Practically, the loss function can be implemented in two forms, named pre-average and post-average, as extended in Eq. (19 - 20) respectively.

$$\mathcal{L}_{F',\text{pre}} = \lambda_t \|\nabla_{F'_t} \log q'(F'_t|F'_0) - \text{Mean}(\hat{\epsilon}'_F)\|_2^2, \tag{19}$$

$$\mathcal{L}_{F',\text{post}} = \lambda_t \text{Mean}\big(\|\nabla_{F'_t} \log q'(F'_t|F'_0) - \hat{\epsilon}'_F\|_2^2\big). \tag{20}$$

Intuitively, the pre-average loss enforces the average output of each Wyckoff position to match with the label on the base node, while the post-average loss minimizes the $L2$-distances of each atom. Table 4 reveals the superior performance of the post-average model, which we adopt for all subsequent experiments.

**Towards structures with customized symmetries** Our method enables structure generation under given space group constraints, hence allowing the creation of diverse structures from the same composition but based on different space groups. To illustrate the versatility of our approach, we visualize some resulting structures in Figure 3 which demonstrates the distinct structures generated under various space group constraints.

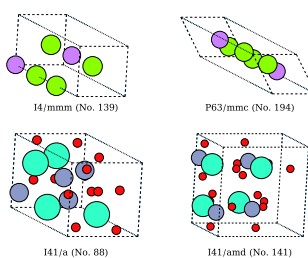

Figure 3: Generation under different space groups.

## 6 CONCLUSION

In this work, we propose DiffCSP++, a diffusion-based approach for crystal generation that effectively incorporates space group constraints. We decompose the complex space group constraints into invariant lattice representations of different crystal families and the symmetric atom types and coordinates according to Wyckoff positions, ensuring compatibility with the backbone model and the diffusion process. Adequate experiments verify the reliability of DiffCSP++ on crystal structure prediction and ab initio generation tasks. Notably, our method facilitates the generation of structures from specific space groups, opening up new opportunities for material design, particularly in applications where certain space groups or templates are known to exhibit desirable properties.

ACKNOWLEDGMENTS

This work is supported by the National Science and Technology Major Project under Grant 2020AAA0107300, the National Natural Science Foundation of China (No. 61925601, 62376276), Beijing Nova Program (20230484278), and Alibaba Damo Research Fund.

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

## A THEORETICAL ANALYSIS

### A.1 PROOF OF PROPOSITION 1

The proposition 1 is rewritten and proved as follows.

**Proposition 1** (Polar Decomposition). *An invertible matrix $\boldsymbol{L} \in \mathbb{R}^{3\times3}$ can be uniquely decomposed into $\boldsymbol{L} = \boldsymbol{Q}\exp(\boldsymbol{S})$, where $\boldsymbol{Q} \in \mathbb{R}^{3\times3}$ is an orthogonal matrix, $\boldsymbol{S} \in \mathbb{R}^{3\times3}$ is a symmetric matrix and $\exp(\boldsymbol{S}) = \sum_{n=0}^{\infty} \frac{\boldsymbol{S}^n}{n!}$ defines the exponential mapping of $\boldsymbol{S}$.*

*Proof.* Given an invertible matrix $\boldsymbol{L} \in \mathbb{R}^{3\times3}$, we first calculate the inner product term $\boldsymbol{J} = \boldsymbol{L}^\top\boldsymbol{L}$. As $\boldsymbol{J}$ is symmetric, we can formulate its eigendecomposition as $\boldsymbol{J} = \boldsymbol{U}\boldsymbol{\Lambda}\boldsymbol{U}^\top$, where $\boldsymbol{U} \in \mathbb{R}^{3\times3}$ is the square matrix composed by eigenvectors of $\boldsymbol{J}$ and $\boldsymbol{\Lambda} \in \mathbb{R}^{3\times3}$ is a diagonal matrix with eigenvalues of $\boldsymbol{J}$ as diagonal elements. The required symmetric matrix can be achieved by $\boldsymbol{S} = \frac{1}{2}\boldsymbol{U}\log(\boldsymbol{\Lambda})\boldsymbol{U}^\top$.

As $\boldsymbol{S}$ is obviously symmetric, we need to prove that $\boldsymbol{Q} = \boldsymbol{L}\exp(\boldsymbol{S})^{-1}$ is orthogonal, *i.e.* $\boldsymbol{Q}^\top\boldsymbol{Q} = \boldsymbol{I}$. To see this, we have

$$
\begin{aligned}
\boldsymbol{Q}^\top\boldsymbol{Q} &= \boldsymbol{Q}^\top\boldsymbol{L}\exp(\boldsymbol{S})^{-1} \\
&= \boldsymbol{Q}^\top\boldsymbol{L}\exp\big(\frac{1}{2}\boldsymbol{U}\log(\boldsymbol{\Lambda})\boldsymbol{U}^\top\big)^{-1} \\
&= \boldsymbol{Q}^\top\boldsymbol{L}\boldsymbol{U}\exp\big(-\frac{1}{2}\log(\boldsymbol{\Lambda})\big)\boldsymbol{U}^\top \\
&= \boldsymbol{Q}^\top\boldsymbol{L}\boldsymbol{U}\sqrt{\boldsymbol{\Lambda}^{-1}}\boldsymbol{U}^\top \\
&= \big(\boldsymbol{L}\boldsymbol{U}\sqrt{\boldsymbol{\Lambda}^{-1}}\boldsymbol{U}^\top\big)^\top\boldsymbol{L}\boldsymbol{U}\sqrt{\boldsymbol{\Lambda}^{-1}}\boldsymbol{U}^\top \\
&= \boldsymbol{U}\sqrt{\boldsymbol{\Lambda}^{-1}}\boldsymbol{U}^\top\boldsymbol{L}^\top\boldsymbol{L}\boldsymbol{U}\sqrt{\boldsymbol{\Lambda}^{-1}}\boldsymbol{U}^\top \\
&= \boldsymbol{U}\sqrt{\boldsymbol{\Lambda}^{-1}}\boldsymbol{U}^\top\boldsymbol{U}\boldsymbol{\Lambda}\boldsymbol{U}^\top\boldsymbol{U}\sqrt{\boldsymbol{\Lambda}^{-1}}\boldsymbol{U}^\top \\
&= \boldsymbol{U}\sqrt{\boldsymbol{\Lambda}^{-1}}\boldsymbol{\Lambda}\sqrt{\boldsymbol{\Lambda}^{-1}}\boldsymbol{U}^\top \\
&= \boldsymbol{U}\boldsymbol{U}^\top \\
&= \boldsymbol{I}.
\end{aligned}
$$

From the above construction, we further have that the decomposition is unique, as $\exp(\boldsymbol{S})$ is positive definite. $\qquad\square$

### A.2 PROOF OF PROPOSITION 2

We begin with the following definition.

**Definition 1** (Frobenius Inner Product in Real Space). *Given $\boldsymbol{A}, \boldsymbol{B} \in \mathbb{R}^{3\times3}$, the Frobenius inner product is defined as $\langle\boldsymbol{A}, \boldsymbol{B}\rangle_F = tr(A^\top B)$, where $tr(\cdot)$ denotes the trace of the matrix.*

The proposition 2 is rewritten as follows.

**Proposition 2.** $\forall \boldsymbol{S} \in \mathbb{R}^{3\times3}, \boldsymbol{S} = \boldsymbol{S}^\top, \exists \boldsymbol{k} = (k_1, \cdots, k_6), s.t. \boldsymbol{S} = \sum_{i=1}^{6} k_i\boldsymbol{B}_i.$

*Proof.* Based on the above definition, we can easily find that $\langle\boldsymbol{B}_i, \boldsymbol{B}_j\rangle_F = 0, \forall i, j = 1, \cdots, 6$ and $i \neq j$, meaning that the bases defined in § 4.1 are orthogonal bases, and the coefficients of the linear combination $\boldsymbol{S} = \sum_{i=1}^{6} k_i\boldsymbol{B}_i$ can be formed as

$$
k_i = \frac{\langle\boldsymbol{S}, \boldsymbol{B}_i\rangle_F}{\sqrt{\langle\boldsymbol{B}_i, \boldsymbol{B}_i\rangle_F}}. \tag{21}
$$

As the 3D symmetric matrix $\boldsymbol{S} \in \mathbb{R}^{3\times3}$ has 6 degrees of freedom (Larson, 2016), it can be uniquely represented by the coefficient vector $\boldsymbol{k} = (k_1, \cdots, k_6)$. $\qquad\square$

A.3 CONSTRAINTS FROM DIFFERENT CRYSTAL FAMILIES

In Crystallography, the lattice matrix $\boldsymbol{L} = [\boldsymbol{l}_1, \boldsymbol{l}_2, \boldsymbol{l}_3]$ can also be represented by the lengths $a, b, c$ and angles $\alpha, \beta, \gamma$ of the parallelepiped. Specifically, we have

$$
\begin{cases}
a = \|\boldsymbol{l}_1\|_2, \\
b = \|\boldsymbol{l}_2\|_2, \\
c = \|\boldsymbol{l}_3\|_2, \\
\alpha = \arccos \frac{\langle \boldsymbol{l}_2, \boldsymbol{l}_3 \rangle}{b \cdot c}, \\
\beta = \arccos \frac{\langle \boldsymbol{l}_1, \boldsymbol{l}_3 \rangle}{a \cdot c}, \\
\gamma = \arccos \frac{\langle \boldsymbol{l}_1, \boldsymbol{l}_2 \rangle}{a \cdot b}.
\end{cases}
\tag{22}
$$

Based on such notation, the inner product matrix $\boldsymbol{J}$ can be further formulated as

$$
\boldsymbol{J} = \boldsymbol{L}^\top \boldsymbol{L} = \begin{bmatrix} a^2 & ab \cos \gamma & ac \cos \beta \\ ab \cos \gamma & b^2 & bc \cos \alpha \\ ac \cos \beta & bc \cos \alpha & c^2 \end{bmatrix} = \boldsymbol{U} \boldsymbol{\Lambda} \boldsymbol{U}^\top.
\tag{23}
$$

Moreover, according to Appendix A.2, we can formulate the corresponding matrix in the logarithmic space as

$$
\boldsymbol{S} = \sum_{i=1}^{6} k_i \boldsymbol{B}_i = k_6 \boldsymbol{I} + \begin{bmatrix} k_4 + k_5 & k_1 & k_2 \\ k_1 & k_5 - k_4 & k_3 \\ k_2 & k_3 & -2k_5 \end{bmatrix} = \frac{1}{2} \boldsymbol{U} \log(\boldsymbol{\Lambda}) \boldsymbol{U}^\top.
\tag{24}
$$

We will specify the cases of the 6 crystal families separately as follows. Note that different with previous works which directly applies constraints on lattice parameters (Liu et al., 2023), we focus on the constraints in the logrithmic space, which plays a significant role in designing the diffusion process.

**Triclinic.** As discussed in Appendix A.2, a triclinic lattice formed by an arbitrary invertible lattice matrix can be represented as the linear combination of 6 bases under no constraints.

**Monoclinic.** Monoclinic lattices require $\alpha = \gamma = 90°$, where $\boldsymbol{J}$ can be simplified as

$$
\boldsymbol{J}_M = \begin{bmatrix} a^2 & 0 & ac \cos \beta \\ 0 & b^2 & 0 \\ ac \cos \beta & 0 & c^2 \end{bmatrix}
\tag{25}
$$

Obviously, $\boldsymbol{J}_M$ has an eigenvector $\boldsymbol{e}_2 = (0, 1, 0)^\top$ as $\boldsymbol{J}_M \boldsymbol{e}_2 = b^2 \boldsymbol{e}_2$. As $\boldsymbol{J}$ and $\boldsymbol{S}$ have the same eigenvectors, we have $\boldsymbol{S}_M \boldsymbol{e}_2 = (k_1, k_5 + k_6 - k_4, k_3)^\top = \lambda_M \boldsymbol{e}_2$ for some $\lambda_M$. Hence we directly have $\lambda_M = k_5 + k_6 - k_4$ and $k_1 = k_3 = 0$.

**Orthorhombic.** Orthorhombic lattices require $\alpha = \beta = \gamma = 90°$, where we have

$$
\boldsymbol{J}_O = \begin{bmatrix} a^2 & 0 & 0 \\ 0 & b^2 & 0 \\ 0 & 0 & c^2 \end{bmatrix}.
\tag{26}
$$

As $\boldsymbol{S}_O = \frac{1}{2} \log(\boldsymbol{J}_O) = diag(\log(a), \log(b), \log(c))$, we can directly achieve the solution of $\boldsymbol{k}$ as

$$
\begin{cases}
k_1 = k_2 = k_3 = 0, \\
k_4 = \log(a/b)/2, \\
k_5 = \log(ab/c^2)/6, \\
k_6 = \log(abc)/3.
\end{cases}
\tag{27}
$$

**Tetragonal.** Tetragonal lattices have higher symmetry than orthorhombic lattices with $a = b$. We have $k_4 = 0$ by substituting $a = b$ into Eq. (27).

**Hexagonal.** Hexagonal lattices are constrained by $\alpha = \beta = 90°, \gamma = 120°, a = b$, which formulate $\boldsymbol{J}$ as

$$\boldsymbol{J}_H = \begin{bmatrix} a^2 & -\frac{1}{2}a^2 & 0 \\ -\frac{1}{2}a^2 & a^2 & 0 \\ 0 & 0 & c^2 \end{bmatrix} = \begin{bmatrix} -\frac{\sqrt{2}}{2} & \frac{\sqrt{2}}{2} & 0 \\ \frac{\sqrt{2}}{2} & \frac{\sqrt{2}}{2} & 0 \\ 0 & 0 & 1 \end{bmatrix} \begin{bmatrix} \frac{3}{2}a^2 & 0 & 0 \\ 0 & \frac{1}{2}a^2 & 0 \\ 0 & 0 & c^2 \end{bmatrix} \begin{bmatrix} -\frac{\sqrt{2}}{2} & \frac{\sqrt{2}}{2} & 0 \\ \frac{\sqrt{2}}{2} & \frac{\sqrt{2}}{2} & 0 \\ 0 & 0 & 1 \end{bmatrix}. \quad (28)$$

And for $\boldsymbol{S}$, we have

$$\boldsymbol{S}_H = \begin{bmatrix} -\frac{\sqrt{2}}{2} & \frac{\sqrt{2}}{2} & 0 \\ \frac{\sqrt{2}}{2} & \frac{\sqrt{2}}{2} & 0 \\ 0 & 0 & 1 \end{bmatrix} \begin{bmatrix} \log(a) + \frac{1}{2}\log(\frac{3}{2}) & 0 & 0 \\ 0 & \log(a) + \frac{1}{2}\log(\frac{1}{2}) & 0 \\ 0 & 0 & \log(c) \end{bmatrix} \begin{bmatrix} -\frac{\sqrt{2}}{2} & \frac{\sqrt{2}}{2} & 0 \\ \frac{\sqrt{2}}{2} & \frac{\sqrt{2}}{2} & 0 \\ 0 & 0 & 1 \end{bmatrix}$$
$$(29)$$

$$= \begin{bmatrix} \log(a) + \frac{1}{4}\log(\frac{3}{4}) & -\frac{1}{4}\log(3) & 0 \\ -\frac{1}{4}\log(3) & \log(a) + \frac{1}{4}\log(\frac{3}{4}) & 0 \\ 0 & 0 & \log(c) \end{bmatrix}. \quad (30)$$

Combine Eq. (24) and Eq. (30), we have the solution as

$$\begin{cases} k_2 = k_3 = k_4 = 0, \\ k_1 = -\log(3)/4, \\ k_5 = \log(\frac{\sqrt{3}a^2}{2c^2})/6, \\ k_6 = \log(\frac{\sqrt{3}}{2}a^2 c)/3. \end{cases} \quad (31)$$

**Cubic.** Cubic lattices extend tetragonal lattices to $a = b = c$, changing the solution in Eq. (27) into $k_1 = k_2 = k_3 = k_4 = k_5 = 0, k_6 = \log(a)$.

# B  IMPLEMENTATION DETAILS

## B.1  ARCHITECTURE OF THE DENOISING BACKBONE

We illustrate the architecture of the model described in § 4.4 in Figure 4. The Fourier coordinate embedding $\psi_{\text{FT}}$ is defined as

$$\psi_{\text{FT}}(\boldsymbol{f})[c, k] = \begin{cases} \sin(2\pi m f_c), k = 2m, \\ \cos(2\pi m f_c), k = 2m + 1. \end{cases} \quad (32)$$

## B.2  COMBINATION WITH SUBSTITUTION-BASED ALGORITHMS

Crystal structure prediction (CSP) requires predicting the crystal structure from the given composition. To conduct our method on the CSP task, we must initially select an appropriate space group and assign each atom a Wyckoff position. We achieve this goal via a substitution-based method, CSPML (Kusaba et al., 2022), which first retrieves a template structure from the training set according to the query composition, and then substitutes elements in the template with those of the query. We depict the prediction pipeline in Figure 5, including the following steps.

**Template Retrieval.** Given a composition as a query, CSPML initially identifies all structures within the training set that share the same compositional ratio (for instance, 1:1:3 for $CaTiO_3$). The retrieved candidates are then ranked using a model based on metric learning. This model is trained on pairwise data derived from the training set. For structures $\mathcal{M}_i, \mathcal{M}_j$, we obtain compositional fingerprints $Fp(c, i), Fp(c, j)$ via XenonPy (Liu et al., 2021) and structural fingerprints $Fp(s, i), Fp(s, j)$ via CrystalNN (Zimmermann & Jain, 2020). The model $\phi$ is trained via the binary classification loss

$$\mathcal{L}_{CSPML} = BCE(\phi(|Fp_{c,i} - Fp(c, j)|), \mathbf{1}_{\|Fp(s,i) - Fp(s,j)\| < \delta}).$$

Here, $\delta$ is a threshold used to determine if the structures of $\mathcal{M}_i$ and $\mathcal{M}_j$ are similar. We adopt $\delta = 0.3$ in line with the setting of Kusaba et al. (2022). The ranking score is defined as $\phi(|Fp_{(c}, q) -$

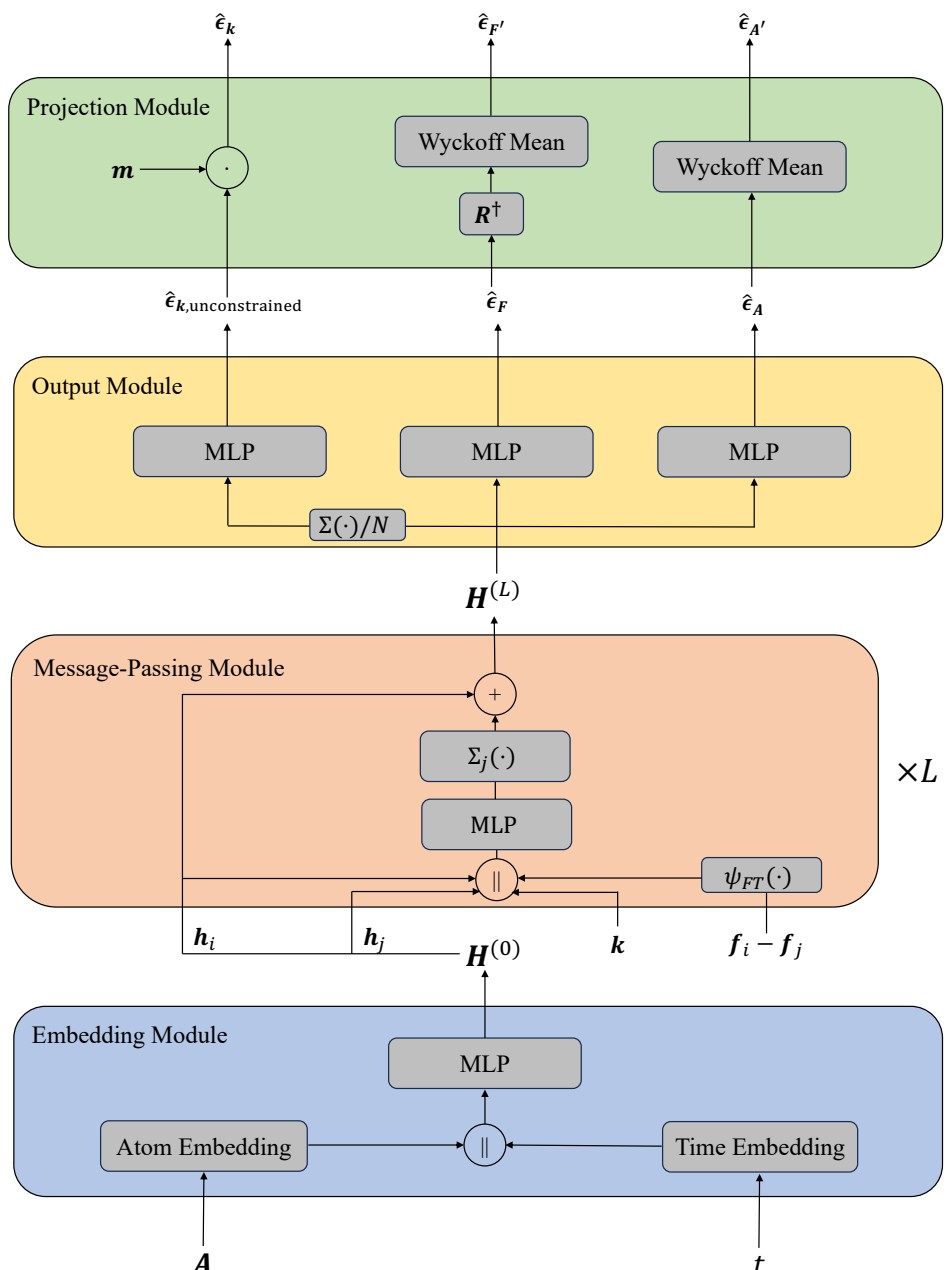

Figure 4: Architecture of the denoising model.

$Fp(c, k)|)$ for a query composition $A_q$ and a candidate composition $A_k$, implying the probability of similarity.

**Element Substitution.** The second step is to assign the atoms in the query composition to the template with the corresponding element ratio (1/5 for Ca in $CaTiO_3$). For the elements with the same ratio (Ca and Ti), we solve the optimal transport with the $\mathcal{L}_2$-distance between the element descriptors as the cost.

**Refinement.** Finally, we refine the structure via DiffCSP++ by adding noise to timestep $t$ and apply the generation process under the constraints provided by the template. Practically, we select $t = 50$ for MPTS-52 and $t = 100$ for Perov-5 and MP-20.

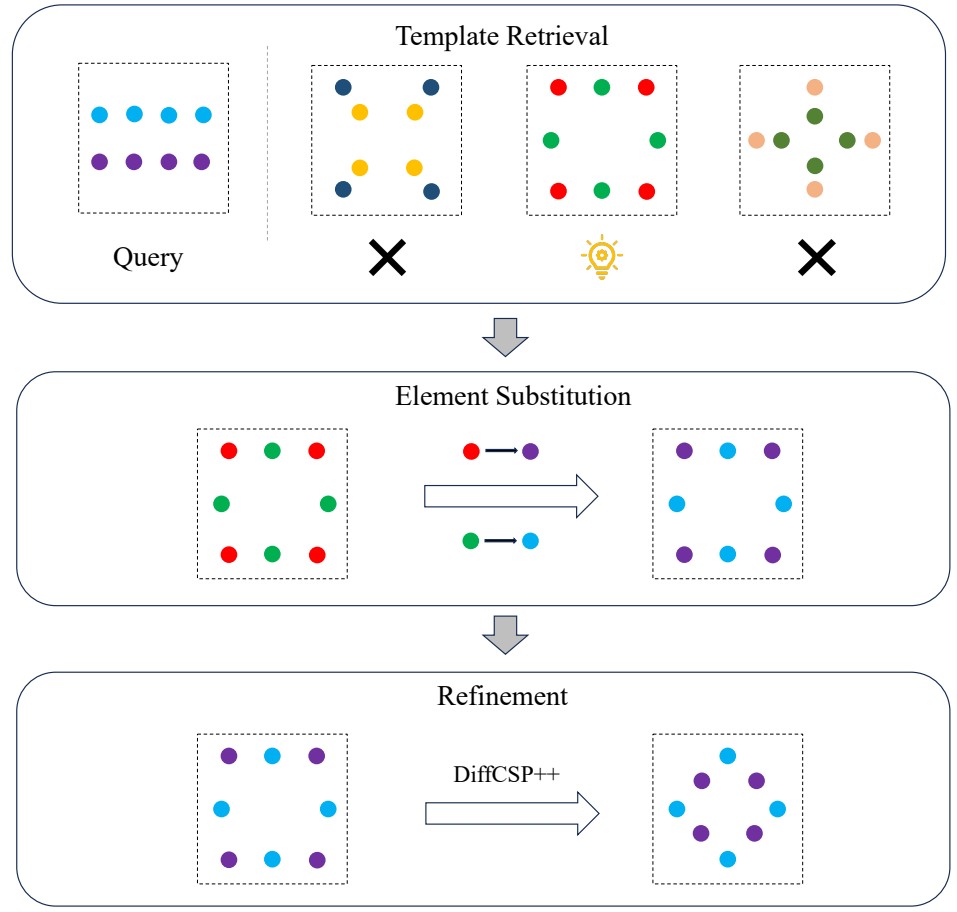

Figure 5: Pipeline of DiffCSP++ combined with CSPML.

Table 5: CSP results of the CSPML templates. MR stands for Match Rate.

|  | Perov-5 | | MP-20 | | MPTS-52 | |
| --- | --- | --- | --- | --- | --- | --- |
|  | MR (%) | RMSE | MR (%) | RMSE | MR (%) | RMSE |
| CSPML (Kusaba et al., 2022) | 51.84 | 0.1066 | 70.51 | 0.0338 | 36.98 | **0.0664** |
| DiffCSP++ (w/ CSPML) | **52.17** | **0.0841** | **70.58** | **0.0272** | **37.17** | 0.0676 |

**More Results.** We further provide the performance of the CSPML templates in Table 5. DiffCSP++ exhibits generally higher match rates and lower RMSE values upon the CSPML templates. This underscores the model's proficiency in refining structures. Note that the refinement step is independent of the template-finding method, and more powerful ranking models or substitution algorithms may further enhance the CSP performance.

### B.3 HYPER-PARAMETERS AND TRAINING DETAILS

We follow the same data split as proposed in CDVAE (Xie et al., 2021) and DiffCSP (Jiao et al., 2023). For the implementation of the CSPML ranking models, we construct 100,000 positive and 100,000 negative pairs from the training set for each dataset to train a 3-layer MLP with 100 epochs and a $1 \times 10^{-3}$ learning rate. To train the DiffCSP++ models, we train a denoising model with 6 layers, 512 hidden states, and 128 Fourier embeddings for each task and the training epochs are set to 3500, 4000, 1000, 1000 for Perov-5, Carbon-24, MP-20, and MPTS-52. The diffusion step is set to $T = 1000$. We utilize the cosine scheduler with $s = 0.008$ to control the variance of the DDPM process on $\boldsymbol{k}$ and $\boldsymbol{A}$, and an exponential scheduler with $\sigma_1 = 0.005, \sigma_T = 0.5$ to

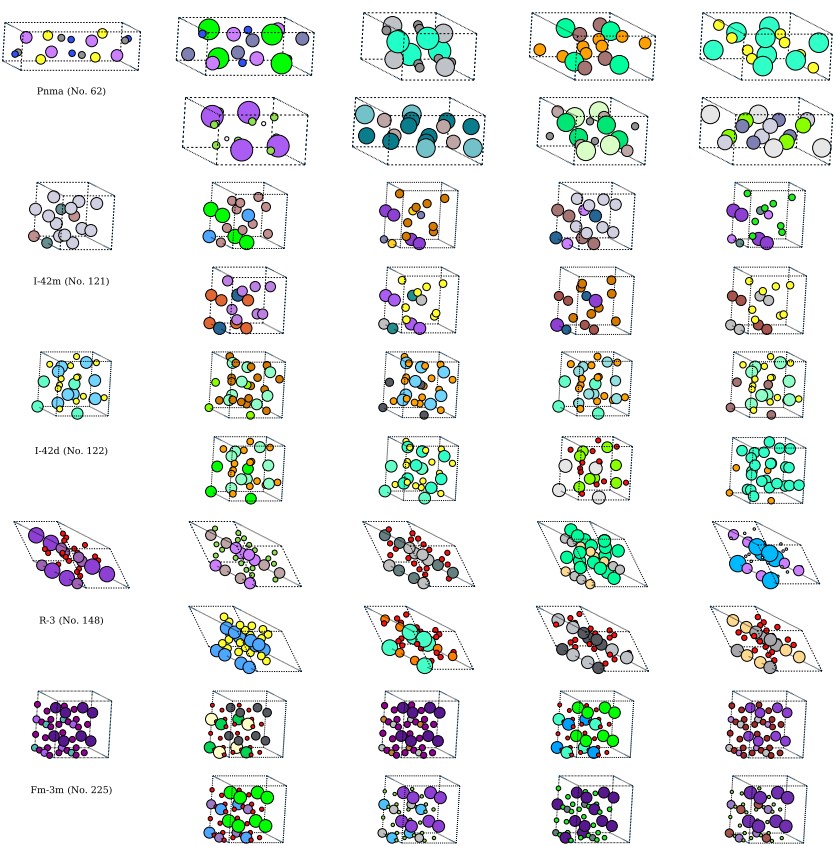

Figure 6: Different structures generated upon the same space group constraints.

control the noise scale on $\boldsymbol{F}$. The loss coefficients are set as $\lambda_{\boldsymbol{k}} = \lambda'_{\boldsymbol{F}} = 1, \lambda'_{\boldsymbol{A}} = 20$. We apply $\gamma = 2 \times 10^{-5}$ for Carbon-24, $1 \times 10^{-5}$ for MPTS-52 and $5 \times 10^{-6}$ for other datasets for the corrector steps during generation. For sampling from $q'$ in Eq. (7), we first sample $\boldsymbol{\epsilon_F} \sim \mathcal{N}(0, \sigma_t^2 \boldsymbol{I})$, select $\boldsymbol{R}_{s_0}$ for each Wyckoff position to acquire $\boldsymbol{\epsilon'_{F'}}[:, s] = \boldsymbol{R}^{\dagger}_{s_0} \boldsymbol{\epsilon_{F'}}[:, s]$, and finally achieve $\boldsymbol{F}'_t$ as $\boldsymbol{F}'_t = w(\boldsymbol{F}'_0 + \boldsymbol{\epsilon'_{F'}})$, where the operation $w(\cdot)$ preserves the fractional part of the input coordinates. To expand the atom types and coordinates of $N'$ Wyckoff positions to $N$ atoms, we first ensure that all atoms in one Wyckoff position have the same type, *i.e.* $\boldsymbol{a}_{s_i} = \boldsymbol{a}'_s$, and then determine the fractional coordinate of each atom via the basic fractional coordinate $\boldsymbol{f}'_s$ and the corresponding transformation pair $(\boldsymbol{R}_{s_i}, \boldsymbol{t}_{s_i})$, meaning $\boldsymbol{f}_{s_i} = \boldsymbol{R}_{s_i} \boldsymbol{f}'_s + \boldsymbol{t}_{s_i}$.

## C  MORE VISUALIZATIONS

We provide visualizations in § 5.4 from a CSP perspective to demonstrate the proficiency of our method in generating structures of identical composition but within varying space groups. Transitioning to the ab initio generation task, we attain an inverse objective, that is, to generate diverse structures originating from the same space group as determined by the template structure. This is further illustrated in Figure 6.

## D  CODES

Our code is available at `https://github.com/jiaor17/DiffCSP-PP`.

