# OpenReview forum: "Space Group Constrained Crystal Generation"
_ICLR.cc/2024/Conference — ICLR 2024 poster_

### Official Review · Reviewer_34Zd · 2023-10-27

**Soundness:** 3 good
**Presentation:** 3 good
**Contribution:** 3 good
**Rating:** 6
**Confidence:** 3

**Summary:**

This paper proposed to consider crystal symmetry group constraints during crystal structure prediction, and proposed decomposition technic to control the symmetry group for lattice and fractional coordinates of atoms in the cell. The idea is interesting and the performances beyond DiffCSP is reasonable and significant.

**Strengths:**

- The first crystal structure prediction method consider crystal symmetry groups.
- SOTA performances beyond DiffCSP.
- Interesting use of crystal symmetries, with controls that can be used during generation.

**Weaknesses:**

- The model details are not provided, some details about the model are missing. This makes me a little confused about what kind of model they are using, e.g., invariant ones or Equivariant ones.
- How they construct the crystal structures from reduced number of atoms N' is not clear.

**Questions:**

As listed in the weaknesses.

---

> ### Author Response · Authors · 2023-11-18
> **Response to Reviewer 34Zd**
>
> Thanks for your suggestions! We provide the following responses to your concerns:
>
> > **W1: The model details are not provided, some details about the model are missing. This makes me a little confused about what kind of model they are using, e.g., invariant ones or Equivariant ones.**
>
> Thank you for raising this concern. Our denoising model is invariant to O(3) transformations on lattices, as we denoise the invariant representation $\mathbf{k}$ rather than $\mathbf{L}$. Additionally, the model is translation invariant with respect to fractional coordinates, since it takes the **relative** coordinates as input, which remain unchanged under translation.
>
> We have further improved the clarity of our model description in Section 4.4 and provided a more detailed illustration of the model architecture in Figure 4, Appendix B.1. Specifically, our denoising model consists of four main modules:
> - **Embedding Module** combines the atom embedding and the sinusoidal time embedding into an MLP to acquire the initial node feature.
> - **Message-Passing Module** in Eq. (12-13) iteratively updates the node feature $\mathbf{h}_i$ via the invariant representation $\mathbf{k}$ of the lattice, the node feature $\mathbf{h}_j$ and the relative fractional difference $\mathbf{f}_j - \mathbf{f}_i$ between all other nodes.
> - **Output Module** in Eq. (14-15) yields the denoising terms without considering the space group constraints.
> - **Projection Module** finally enforces the constraints by projecting the lattice onto a specific crystal family in Eq. (16) and aligning the atoms belonging to the same Wyckoff position in Eq. (17-18).
>
> > **W2: How they construct the crystal structures from reduced number of atoms N' is not clear.**
>
> Sorry for the unclearity. Given a crystal structure with $N'$ Wyckoff positions, we denote that each Wyckoff position contains $n_s$ atoms satisfying $1\leq s\leq N'$. Here, $N=\sum_{s=1}^{N'}n_s$ represents the total number of atoms in a unit cell. We construct the whole crystal structures from reduced number of atoms $N'$ in this way:
> 1) For each $s$-th Wyckoff position, we use the notations $\mathbf{a}'_s$ and $\mathbf{f}'_s$ to respectively represent the atom type and basic fractional coordinate of  Wyckoff position.
> 2) For each atom $s_i$ in the $s$-th Wyckoff position, we denote the atom type and fractional coordinate as $\mathbf{a}\_{s\_i}, \mathbf{f}\_{s\_i}$ (note that $1\leq s_i \leq n_s$).
> 5) First, we ensure that all atoms in one Wyckoff position have the same type, meaning $\mathbf{a}\_{s\_i}=\mathbf{a}_{s}$.
> 6) Next, for each atom $s_i$, we determine its fractional coordinate using the basic fractional coordinate and its corresponding transformation pair $(\mathbf{R}\_{s_i},\mathbf{t}\_{s_i})$. Specifically, we calculate $\mathbf{f}\_{s\_i}$ by $\mathbf{f}\_{s_i}=\mathbf{R}\_{s_i}\mathbf{f}'\_s+\mathbf{t}\_{s_i}$.
> 7) After iterating over all atoms and all Wyckoff positions, we finally obtain the the whole crystal structures.
>
> The above explanations have been added into Appendix B.3 to address the reviewer's concern. Once again, we appreciate the reviewer for the positive recognition of our work and hope the reviewer's concerns regarding clarity are well-addressed.

---

> ### Author Response · Authors · 2023-11-21
> **Looking Forward to Your Feedback**
>
> Dear Reviewer 34Zd,
>
> We greatly appreciate your thoughtful insights and valuable comments on our work. As the reviewer-author discussion period is ending, we would like to kindly remind you that we are looking forward to your feedback on our response.
>
>
> We have provided more details about the architecture of the denoising model, and the relationship between the Wyckoff positions and individual atoms. If you have any further questions on our work, please do not hesitate to let us know.
>
> Best regards,
>
> Authors

---

> > ### Comment · Reviewer_34Zd · 2023-11-22
> > **Follow up question**
> >
> > Thank you for the information.
> >
> > I am curious how to predict F or your mentioned fractional difference using an invariant network, these outputs are Equivariant.

---

> > > ### Author Response · Authors · 2023-11-23
> > > **Response to the Follow up question**
> > >
> > > > **I am curious how to predict F or your mentioned fractional difference using an invariant network, these outputs are Equivariant.**
> > >
> > > Thanks for your interest! Yes, the invariant denoising network will yield the equivariant update of the fractional coordination $\mathbf{F}$. We provide the detailed explanations below.
> > >
> > > We employ the predictor-corrector sampler, as used in DiffCSP [A], to equivariantly generate the fractional coordinates $\mathbf{F}\_{t-1}$ from $\mathbf{F}\_t$ via the invariant denoising model $\hat{\epsilon}\_{\mathbf{F}}(\mathbf{F}\_t,t)$. The process consists of the following steps:
> > >
> > > Predictor Step: $\mathbf{F}\_{t-0.5}=w(\mathbf{F}\_{t}+(\sigma\_t^2-\sigma\_{t-1}^2)\hat{\epsilon}\_{\mathbf{F}}(\mathbf{F}\_t,t)+\frac{\sigma\_{t-1}}{\sigma\_t}\sqrt{\sigma\_t^2-\sigma\_{t-1}^2}\epsilon_P),$
> > >
> > > Corrector Step: $\mathbf{F}\_{t-1}=w(\mathbf{F}\_{t-0.5}+\gamma\frac{\sigma\_{t-1}}{\sigma_1}\hat{\epsilon}\_{\mathbf{F}}(\mathbf{F}\_{t-0.5},t-1)+\sqrt{2\gamma\frac{\sigma\_{t-1}}{\sigma\_1}}\epsilon_C)$，
> > >
> > > where the function $w(\cdot)$ returns the fractional part of each element of the input.
> > >
> > > Note that the denoising term $\hat{\epsilon}\_{\mathbf{F}}$ is periodic translation **invariant**, as already explained in our previous responses. This is because $\hat{\epsilon}\_{\mathbf{F}}(w(\mathbf{F}\_t+\mathbf{r1}^\top),t)=\hat{\epsilon}\_{\mathbf{F}}(\mathbf{F}\_t,t)$ holds true for any translation $\mathbf{r}$. Consequently, this leads to a periodic translation **equivariant** generation process. To illustrate this, we can consider the mean value of the predictor step $\mu\_{t-0.5}(\mathbf{F}\_t)=w(\mathbf{F}\_{t}+(\sigma_t^2-\sigma_{t-1}^2)\hat{\epsilon}\_{\mathbf{F}}(\mathbf{F}\_t,t))$, which yields:
> > >
> > > $$
> > > \begin{align}
> > > \mu\_{t-0.5}(w(\mathbf{F}\_t+\mathbf{r1}^\top))&=w(\mathbf{F}\_{t}+\mathbf{r1}^\top+(\sigma_t^2-\sigma_{t-1}^2)\hat{\epsilon}\_{\mathbf{F}}(w(\mathbf{F}\_t+\mathbf{r1}^\top),t)) \\\\
> > > &=w(\mathbf{F}\_{t}+\mathbf{r1}^\top+(\sigma\_t^2-\sigma\_{t-1}^2)\hat{\epsilon}\_{\mathbf{F}}(\mathbf{F}\_t,t)) \\\\
> > > &=w(\mu\_{t-0.5}(\mathbf{F}\_t)+\mathbf{r1}^\top).
> > > \end{align}
> > > $$
> > >
> > > It means the periodic translation of the input leads to the periodic translation of the output, hence ensuring periodic translation equivariance. Similar derivation can be provided for the corrector step.
> > >
> > >
> > > We hope this explanation addresses your question and clarifies the equivariant nature of the process.
> > >
> > > [A] Jiao, Rui, et al. "Crystal structure prediction by joint equivariant diffusion." arXiv preprint arXiv:2309.04475 (2023).

---

### Official Review · Reviewer_GFzi · 2023-11-01

**Soundness:** 4 excellent
**Presentation:** 3 good
**Contribution:** 4 excellent
**Rating:** 8
**Confidence:** 3

**Summary:**

A novel approach for crystal structure generation and ab-initio
crystal generation. The approach complements a recent solution based
on diffusion models with the support for space group
constraints. Experimental results confirm the potential of the
proposed solution.

**Strengths:**

Tackling a relevant application which requires non-trivial changes to existing solutions.

A sound and well-motivated approach.

Well-written and well organized manuscript.

**Weaknesses:**

The relationship of the method with DiffCSP, on which it builds,
should be better clarified, so as to more clearly highlight the
contribution of the space group constraint.

I am not entirely happy with how results are reported in Table 2. My
understanding is that given that the space group of the GT data is
typically not available, the real method is what you call DiffCSP++
(w/ CSPML) (which I would name DiffCSP++), and this is the method you
should compare with the competitors (with boldface for best performing
method etc).  The approach using GT space group (that you name
DiffCSP++) is an upper bound on the achievable performance, and should
be reported as such (including the discussion on future work about
better template-finding algorithm, that you already have in the paper).

This also affect table 3, in case DiffCSP++ uses GT space group and
this is not a kind of knowledge to be expected in ab-initio generation
(I do not know the domain).

**Questions:**

Does DiffCSP++ in Table 3 use GT space group? is this to be expected?

Also, see weaknesses for clarification requests.

---

> ### Author Response · Authors · 2023-11-18
> **Response to Reviewer GFzi**
>
> Thanks for your constructive comments! We provide more explanations to address your concerns as follows.
>
> > **W1: The relationship of the method with DiffCSP, on which it builds, should be better clarified, so as to more clearly highlight the contribution of the space group constraint.**
>
> Thanks for your suggestions, and we have further highlight the difference from DiffCSP in the revised version. Here, we summarize the difference in two main points:
> 1) DiffCSP directly generates the lattice matrix $\mathbf{L}$, while this paper instead generates the coefficients $k$ of the basis in Eq. (12) and (14) to better fulfil the space group constraint. Moreover, according to Proposition 1, the correspondence between the space of the coefficients $k$ and that of the inner product $\mathbf{L}^\top\mathbf{L}$ is bijective. This means our model enjoys the same expressivity with DiffCSP. This is supported by the ablation study in Table 4 where it showcased the comparable performance between DiffCSP and DiffCSP-k.
> 2) In Eq. (17), we first project the denoising term in the subspace of Wyckoff position via the pseudo-inverse of the matrix $\mathbf{R}$ of each atom, and then compute the mean of the projected denoising terms of the atoms in the same Wyckoff position. This constraint is not considered in the original DiffCSP but is crucial for adapting to the diffusion framework described in Section 4.3.
>
> > **W2: The presentation of Table 2.**
>
> Thanks for your comment. We agree that the results of DiffCSP++ with GT space groups, which are typically unavailable in real-world settings and should be reported as an upper bound on achievable performance. We have reflected this point in Table 2 by renaming DiffCSP++ as DiffCSP++(w/ GT) to avoide the confusion in the revised paper.
>
> > **W3 & Q1: Does DiffCSP++ in Table 3 use GT space group? is this to be expected?**
>
> No, DiffCSP++ in Table 3 does NOT use GT space group. Sorry for the confusion. Table 3 is for the ab initio generation task, where the atom types, coordinates and the entire lattice matrix are jointly generated. For the generation of each crystal, we randomly select a sample **from the training set** as the template for the space group. Note that we did no apply GT space group in this task, as the generation is not based on the templates **from the testing set**. Therefore, the results in Table 3 are consistent with the problem domain, and our method can be fairly compared with other approaches and applicable in the real-world setting.

---

> ### Author Response · Authors · 2023-11-21
> **Looking Forward to Your Feedback**
>
> Dear Reviewer GFzi,
>
> Thanks again for your insightful comments, and this is a kind reminder that as the reviewer-author discussion period is ending soon, we are looking forward to your feedback on our response.
>
> In our response and the latest revision, we have elaborated on the distinctions between our proposed method and the original DiffCSP, as well as clarified the application of GT. Thank you once again for your time and expertise. Please let us know if you have any further questions.
>
> Best regards,
>
> Authors

---

### Official Review · Reviewer_GbgC · 2023-11-10

**Soundness:** 3 good
**Presentation:** 3 good
**Contribution:** 3 good
**Rating:** 8
**Confidence:** 2

**Summary:**

The authors propose DiffSCP++ - building upon prior work DiffSCP (Jiao et al., 2023), to incorporate inductive biases of space group symmetries in a computationally tractable form for crystal generation. The primary goal in this work is to learn to sample (via diffusion) from a conditional distribution (given the finite space group) rather than an unconditional generation combined with E(n) equivariant networks. The authors do this by decomposing into two parts -  constraints based on an orthogonal group invariant exponential subspace and constraints of fractional coordinates. Subsequently, to tackle the crystal generation problem, the authors employ diffusion models to jointly generate the lattice, the fractional coordinates and atom types conditioned on the obtained constraints pertinent to the space group. The authors then present results on 4 different datasets, for two different tasks (crystal structure prediction and ab-initio crystal generation) where their model outperforms the compared baselines


Note to AC: My lower confidence rating is because of lack of expertise with literature relevant to crystal generation.

**Strengths:**

1. Most parts of the paper are well written and easy to comprehend for someone not familiar with crystal generation literature.
2. Employs a conditional generation model - which takes into account the space group constraints rather than treating the generation and the invariances as separate modules.
3. Compares to other recent works, such as PGCGM (Zhao et al., 2023) which also incorporate the affine matrices of the space
group as additional input into a Generative Adversarial Network (GAN) model. The big plus of DiffSCP++ is that it is more widely applicable without being constrained to ternary systems.
4. Strong experimental results in comparison to DiffSCP (out of which the model was built out of) and other baselines in the crystal structure prediction task.

**Weaknesses:**

1. Proposition 1 and 2 - are simple extensions of two known results from linear algebra - why is this presented without citations - to linear algebra books from Strang, Roman, etc.?
2. Novelty is definitely present - but not something completely unexpected and draws and builds upon existing literature

**Questions:**

Please address the weakness.

---

> ### Author Response · Authors · 2023-11-18
> **Response to Reviewer GbgC**
>
> Thanks for your valuable comments! We have revised our paper according to your kind suggestions.
>
> > **W1: Proposition 1 and 2 - are simple extensions of two known results from linear algebra - why is this presented without citations - to linear algebra books from Strang, Roman, etc.?**
>
> Thanks for your suggestion. For Proposition 1, the polar decomposition is well-known and we have added the citation about the polar decomposition in the revised paper. For Proposition 2, the 6 orthogonal bases of $3\times 3$ symmetric matrices are specifically designed by us. Although there could be other forms of orthongoal bases, our paper choose the one that straightfowardly distinguishes between different crystal families, as presented in Table 1. We provided a proof of Proposition 2 in Appendix A.2, and have included a citation in the revised paper to support that the space of $3\times 3$ symmetric matrices has a dimension of 6.
>
> > **W2: Novelty is definitely present - but not something completely unexpected and draws and builds upon existing literature.**
>
> Thanks for recognizing our novelty. Here, we would like to further highlight the challengs by injecting space group into the diffusion process. In crystallology, it is well known that space group is closely related important properties of crystals, hence it will be beneficial if the crystal generation model can spontaneously maintain a given space group constraint. However, this is a non-trivial task, which presents two main challenges:
> 1) As listed in Table 1, the lattice must conform to a specific shape according to the space group.
> 2) Symmetric atoms (namely the atoms in the same orbit) must be updated synchronously, as illustrated by the dashed lines in Figure 2.
>
> To address these challenges, we divide the constraint into two manageable parts: the basis constraint of the lattice matrix and the Wyckoff position constraint of the fractional coordinates, each of which is exactly retained during the generation process. Overall, while it is a natural enhancement of existing literature by further considering space group constraints, the way we implement the constraints is novel and non-straightforward.

---

> > ### Comment · Reviewer_GbgC · 2023-11-21
> > **Acknowledge the rebuttal**
> >
> > Dear Authors,
> >
> > Thank you for the rebuttal and updates to the manuscript. Having gone through all the other reviews as well as your responses, I will stick to my scores of accept (8). Good luck!

---

> > > ### Author Response · Authors · 2023-11-21
> > > **Thank you**
> > >
> > > Thank you for your valuable comments and suggestions that greatly improve our paper!

---

### Public Comment · ~Kaifan_Yang1 · 2023-11-16
**Lack some related works**

We appreciate this great work and the new SOTA in CSP by the authors.
However, the constraints among the space group and lattice constants was firstly proposed and used in crystal structure prediction in previous work, PCVAE [1], which aims to determine the the geometry and symmetry of crystals.

The table.1 in this work and Table.3 in PCVAE are similar.
We believe the two works are highly related and should be mentioned in this article.

*[1] Liu, Ke, et al. "PCVAE: A Physics-informed Neural Network for Determining the Symmetry and Geometry of Crystals." 2023 International Joint Conference on Neural Networks (IJCNN). IEEE, 2023.*

---

> ### Author Response · Authors · 2023-11-18
>
> Hi Kaifan,
>
> Thanks for your interest in our work. We have included a discussion of PCVAE in Section 2 of the revised manuscript. While these two works are related in the part of the lattice constraint, our DiffCSP++ greatly differs from PCVAE in several key aspects:
> 1) **Different Tasks**. Given a space group, PCVAE focses only on predicting the lattice, whereas our DiffCSP++ aims at generating a whole crystal structure including both the lattice and atom coordinates/types. Since our task is more challenging, we require to elaborate a generation model that is capable of capturing the distribution of the crystal structure under the space group constraint. In our paper, we employ the diffusion model owing to its powerful generation ability, other than the VAE model used in PCVAE.
> 2) **Different Implementation of Lattice Constraint**. PCVAE represents the lattice using six lattice parameters, i.e., $a, b, c, \alpha, \beta, \gamma$, and predicts the parameter if its corresponding coefficient $p=1$ (see Table 3 in the PCVAE paper). However, since the edge parameters should be possitive and the angle parameters should fall into the range of $[0, 2\pi]$, it is nontrivial to ensure a valid prediction if we particularly apply a diffusion model whose output range is $[-\infty, \infty]$. Therefore, in our paper, we employs the logarithmic representation $\mathbf{k}$ of the lattice in Table 1. In this way, different lattice constraints correspond to different dimensions of $\mathbf{k}$, and more importantly, the range of $\mathbf{k}$ is $\mathbb{R}^6$ which is can be naturally generated by a diffusion model. We further discuss the relationship between $\mathbf{k}$ and the lattice parameters for each crystal family in Appendix A.3.
> 3) **Atom Coordinate Constraint**. Besides the constraint on lattices, our approach additionaly constrains the atom fractional coordinates with the help of Wyckoff position, providing a more comprehensive treatment of space group constraints.

---

### Author Response · Authors · 2023-11-18
**General Response**

We thank all reviewers for their valuable comments. We have carefully revised our paper, with changes indicated in red text, to address the concerns raised. The key revisions include:

- Expanded discussions on related works and additional citations. (To Reviewer GbgC)
- Further explanations of how DiffCSP++ differs from the original DiffCSP in Section 4.4. (To Reviewer GFzi)
- Correction of the name of our method that utilizes ground truth constraints (have been renamed as "DiffCSP++ (w/ GT)") in Table 2, which serves as an upper bound for our approach. (To Reviewer GFzi)
- Improved clarity of the notations in Section 4.2 to better introduce the connections between Wyckoff positions and each independent atom. (To Reviewer 34Zd)
- Detailed illustration of the denoising model architecture in Appendix B.1. (To Reviewer 34Zd)


We hope that these revisions address the reviewers' concerns and enhance the overall quality of our paper.

---

### Meta-Review · Area_Chair_L8Lu · 2023-12-15

**Metareview:**

The paper simplifies the spacegroup constraint by transforming it into a more manageable formulation for incorporation into the generation process. Specifically, the spacegroup constraint is reformulated into two cases: the basis constraint involving the invariant exponential space of the lattice matrix, and the Wyckoff position constraint relating to fractional coordinates. This translation aims to make the constraint more amenable to manual manipulation within the generation process. The method shows strong empirical performance improving over prior work such as DiffCSP.

 All reviewers are positive about the paper and recommend it to be accepted. There was an external public comment asking for a comparison to prior work. The authors convincingly explained the differences to this work. Overall, I recommend accepting the paper.

**Justification For Why Not Higher Score:**

-

**Justification For Why Not Lower Score:**

-

---

### Decision · Program_Chairs · 2024-01-16

Accept (poster)